# A Scoping Review of Climate Change, Climate-Related Disasters, and Mental Disorders among Children in Low- and Middle-Income Countries

**DOI:** 10.3390/ijerph19052896

**Published:** 2022-03-02

**Authors:** Isobel Sharpe, Colleen M. Davison

**Affiliations:** Department of Public Health Sciences, Queen’s University, Kingston, ON K7L 3N6, Canada; isobel.sharpe@queensu.ca

**Keywords:** scoping review, climate change, climate-related disasters, mental health, mental disorders, children, low and middle income, LMICs

## Abstract

Children, particularly those living in low- and middle-income countries (LMICs), are highly vulnerable to climate change and its impacts. Our main objective was to conduct a scoping literature review to determine how exposure to climate change and climate-related disasters influences the presence of mental disorders among children in LMICs. We also aimed to identify gaps in this area of scholarship. We included studies of children in LMICs that had a climate change or climate-related disaster exposure and mental disorder outcome. Twenty-three studies were included in the final synthesis. Fourteen studies were conducted in China, three in India, two each in Pakistan and the Philippines, and one each in Namibia and Dominica. All studies assessed the association between a climate-related disaster exposure and a mental disorder outcome, while none explored broader climate change-related exposures. Post-traumatic stress disorder (*n* = 21 studies) and depression (*n* = 8 studies) were the most common mental disorder outcomes. There was considerable between-study heterogeneity in terms of sample size, follow-up length, and outcome measurement. Overall, the literature in this area was sparse. Additional high-quality research is required to better understand the impacts of climate-related disasters and climate change on mental disorders within this population to ultimately inform future policies and interventions.

## 1. Introduction

Our climate is currently experiencing a rapid and unsustainable increase in temperature, largely driven by human greenhouse gas emissions. As of 2020, the past six years have been the warmest on record [1]. Global surface temperature has risen approximately 1.2 °C above pre-industrial levels [1], and this increase may reach up to 2 °C by the end of this century [2]. As a result, climate change has been recognized as “the biggest global health threat of the 21st century” [3]. The changing climate has a wide range of impacts on human health, such as increasing transmission of infectious diseases, increasing mortality due to heatwaves and air pollution, as well as contributing to undernutrition driven by reduced crop yields [4].

Climate change also harms human mental health and well-being [5,6]. For one, experiences of climate-related disasters, which are increasing in frequency and severity due to climate change [2], contribute to psychological trauma. For instance, exposure to flooding has been associated with a variety of negative mental health outcomes such as post-traumatic stress disorder (PTSD), depression, anxiety, and overall poor well-being [7]. Broader climate change-related events, such as biodiversity shifts and gradual temperature and sea level increase, also contribute to mental health problems. For example, rising sea levels may lead to forced migration, which contributes to mental health problems associated with overcrowding, poverty, and loss of a sense of place [8,9].

Individuals living in low- and middle-income countries (LMICs) are susceptible to the negative mental health effects of climate change and climate-related disasters. Like the rest of the world, many of these countries are experiencing unfavorable changes in temperature and precipitation patterns, as well as greater frequency and severity of climate-related natural disasters [2,10]. Due to potentially weaker health care systems and more limited financial resources for adequate infrastructure, LMICs may have less capacity for mitigation and adaptation than higher-resource areas [2,11]. In addition, there may be a greater need for mental health support [12,13]. Although individuals and communities will respond to climate-related adversities in their own unique ways, meaning that we cannot assume uniformity across populations, aspects of a broader context can place specific populations at potentially greater risk for negative mental health impacts. For example, LMICs where health systems are not currently providing comprehensive mental health services, or where populations are living in poverty, insecurity, or conflict may be at an overall greater risk for negative mental health outcomes. Several systematic and scoping reviews have identified LMIC-specific evidence supporting the various negative mental health outcomes associated with climate change and climate-related disasters [6,14,15,16].

Children are a specific subpopulation that is affected by climate change and climate-related disasters [17]. Relative to adults, children can be more susceptible to negative environmental changes [17,18,19]. Evidence also suggests that stressful early-life events are frequently associated with poor health- and mental health-related outcomes later in life [20,21,22,23]. Furthermore, throughout their lifetimes, children will be forced to bear the future consequences of our widespread climate change inaction and will be required to adapt and develop solutions to these problems. For instance, based on the current climatic trajectory, Thiery et al. [24] found that children born in 2020 will experience 7-fold more heatwaves than children born in 1960. To better quantify these inequities, UNICEF [17] developed a country-level climate change risk index. The index accounted for children’s level of exposure to climate change-related hazards as well as their vulnerabilities related to essential services such as water and sanitation, education, and nutrition, which often compound to produce an array of negative outcomes. According to the index, approximately half the number of children worldwide live in countries at a level of extremely high risk.

In comparison to adults, relatively little research exists on the mental health effects of climate change and climate-related disasters on children. Some narrative reviews have explored this topic [19,25,26,27]; however, none have captured the full extent of the existing literature. In their recent scoping review of climate change and child health, Helldén and colleagues [28] analyzed 371 records, only 12% of which focused on LMICs and an even smaller proportion on mental health. Systematically capturing relevant literature on climate change, climate-related disasters, and mental health in this context will help to identify research gaps and may eventually inform LMIC- and child-specific adaptation policies and mental health interventions.

In 2010, the Global Burden of Disease study found that mental, neurological, and substance abuse disorders accounted for over 10% of all disability-adjusted life years [29]. Mental disorders are an especially burdensome aspect of mental health, and assessing their presence is highly relevant within the context of climate change and climate-related disasters. Therefore, our main objective was to conduct a scoping literature review to determine how exposure to climate change and climate-related disasters influences the presence of mental disorders among children in LMICs. We also aimed to identify and summarize existing gaps in this area of literature.

## 2. Materials and Methods

We used the scoping review methodology to answer our research question. Scoping reviews are used to systematically map a heterogeneous body of literature [30], and we felt it was appropriate given that the settings and designs of studies investigating climate change, climate-related disasters, and mental disorders tend to vary widely. This review aligned with the Preferred Reporting Items for Systematic reviews and Meta-Analyses extension for Scoping Reviews (PRISMA-ScR) checklist [31] and we followed the same methodological approach as our previous scoping review, which investigated climate change and mental health among adults in LMICs [16]. The review was based on an unpublished protocol.

### 2.1. Eligibility Criteria

We included all primary research published in 2007 or later that explored the relationship between a climate change or climate-related disaster exposure and a mental disorder outcome in LMICs (Table 1). Studies were included if they focused on children, defined as having a study population with a mean age of 18 years or less. This definition corresponded with our previous work on adult-focused populations [16]; we felt it was important to capture studies that primarily included children but also those that may have included a small proportion of adults (given that this situation was common within our area of literature). We included studies with exposures that were either climate-related natural disasters or were explicitly mentioned by the study author as “climate change”. The mental disorder outcomes of interest were anxiety, depression, post-traumatic stress, acute stress, substance use and addiction, bipolar, schizophrenia, suicidal behavior, and non-suicidal self-injury [32]. We included studies that evaluated mental disorders based on Diagnostic and Statistical Manual of Mental Disorders (DSM) or International Classification of Diseases (ICD) criteria. For feasibility purposes, only research written in the English language was included.

### 2.2. Information Sources and Search Strategy

We conducted a comprehensive search of both the academic and grey literature. All searches were set to identify records published between 1 January 2007 and 31 December 2019. In terms of the academic literature, we searched five relevant electronic databases: MEDLINE via Ovid (biomedicine, 1946–present), Embase via Ovid (biomedicine and pharmacology, 1947–present), APA PsycInfo via Ovid (psychological, social, behavioral, and health sciences, 1806–present), Global Health via Ovid (global health, 1910–present), and Sociological Abstracts via ProQuest (sociology, 1952–present). With guidance from a health sciences librarian, the search strategy was developed in MEDLINE and translated to the four other databases (Appendix A). We also performed manual searches of the 2019 volumes of the following journals to capture any studies that were not yet indexed in the electronic databases: International Journal of Environmental Research and Public Health, Global Environmental Change, Climatic Change, Disaster Medicine and Public Health Preparedness, and Environmental Health. In addition, we searched the reference lists of all included studies (backward snowballing). We also checked Google Scholar for any relevant citations of the studies (forward snowballing).

In terms of the grey literature, our search strategy was adapted from Godin et al. [35]. First, a list of relevant organizations was generated and targeted website browsing/searching was performed (Appendix A). The list was generated through the authors’ personal knowledge, as well as through Google searching of relevant keywords. If the website contained a search feature, it was used in combination with general browsing. Second, we searched the following grey literature databases: Google Scholar, OAIster, OpenGrey, The Grey Literature Report, ELDIS, BIOSIS Previews, Center for Research Libraries, Networked Digital Library of Theses and Dissertations, and ProQuest Dissertations & Theses Global (Appendix A). Lastly, the popular search engines Google and DuckDuckGo were searched (Appendix A). The first 100 results of each search were screened.

### 2.3. Screening

First, all search results were compiled and duplicates were removed. For feasibility purposes, a single reviewer performed an initial relevancy screen of all unique records to identify those titles/abstracts clearly unrelated to the study objective (i.e., did not have an exposure related to the environment, did not have an outcome related to mental health, did not take place in a LMIC, or were not primary research). Following the initial screen, two independent reviewers completed a formal assessment of eligibility (Table 1) at the title/abstract level and any irrelevant records were excluded. The remaining records were then assessed by two reviewers at the full-text level to confirm eligibility (Table 1). Any conflicts were discussed between the two reviewers and mutual agreement was reached. All aspects of the study organization and screening were managed using the Covidence systematic review software (https://app.covidence.org/, accessed on 21 December 2020 Veritas Health Innovation, Melbourne, Australia).

### 2.4. Data Collection and Analysis

Data extraction was completed for all included studies. Prior to beginning the extraction process, a comprehensive Microsoft Excel data extraction form was developed, which was piloted on a random sample of five included studies and adjusted accordingly. The form consisted of information on study design, baseline characteristics, and qualitative and quantitative study findings (Appendix A). The final results were presented qualitatively in various descriptions, tables, frequency counts, and diagrams.

## 3. Results

The academic literature search returned 6971 unique records, 4536 of which were excluded through the initial relevancy screen (Figure 1). A total of 2435 records were screened for eligibility by two independent reviewers at the title/abstract level and 2076 were excluded. The remaining 359 were assessed at the full-text level and 338 were excluded. In combination with two additional records identified through backwards and forwards snowballing, 23 studies were included in the final qualitative synthesis. No additional records were identified through the grey literature search.

Of the 23 included studies, 22 were quantitative only and one was mixed methods. Twenty-one studies used a cross-sectional design and the remaining two used a prospective cohort design. The included studies spanned six different LMICs: China (*n* = 14 studies), India (*n* = 3), Pakistan (*n* = 2), Philippines (*n* = 2), Namibia (*n* = 1), and Dominica (*n* = 1). All 23 studies assessed climate-related disaster exposures, while none assessed broader climate change-related exposures. Twelve unique disaster events were identified, including four floods, four typhoons/cyclones/tropical storms, two snowstorms, one rainstorm, and one tornado (Figure 2). 

The most common mental disorder outcome was PTSD, assessed by 21 studies. Depression was assessed by eight studies, anxiety by two studies, and acute stress disorder by one study. One study assessed all DSM disorders. Prevalence estimates ranged from 2% [36,37] to 83% [38] for PTSD and from 2% [36] to 66% [39] for depression. While some studies assessed mental disorders using formal diagnostic interviews, e.g., [36,40,41], others used screening tools with validated cut-off scores to establish an informal diagnosis, e.g., [38,39,42]. The most commonly used screening tools were the Child PTSD Symptom Scale (*n* = 8 studies), the Depression Self-Rating Scale for Children (*n* = 4), and the PTSD Checklist for DSM-5 (*n* = 3).

Study sample sizes ranged from *n* = 64 [43] to *n* = 7038 [37]. Most (70%, *n* = 16) studies had a sample size between 100 and 500. Nineteen studies collected samples from school-based populations and four studies collected samples from community-based populations (one sampled females only). Follow-up length ranged from one week [44] to five years [40] post-exposure and the most common follow-up length was 12 months (*n* = 5 studies). Date of publication ranged from 2007–2019, with the large majority of studies (78%, *n* = 18) published in 2015 or later (Figure 3). Eleven studies were published in 2018 alone, eight of which presented identical or very similar samples with varying lengths of follow-up. All but one of the included studies [45] had partial or full LMIC authorship. The following sections present relevant study findings by country and Table 2 provides a detailed summary of all included studies.

### 3.1. China

Fourteen studies were conducted in China. Of those studies, 11 investigated the same disaster event: a 2016 tornado in Yancheng City. Xu et al. [49] measured post-tornado PTSD and depression among a sample of middle school students, reporting 3 month prevalence estimates of 57.5% and 58.7%, respectively. In addition, the authors found significantly greater odds of both disorders among students with family/friends who were injured by the tornado (PTSD: odds ratio (OR) 1.98, 95% confidence interval (CI) 1.01–3.92; depression: OR 2.05, 95% CI 1.03–4.07). A number of studies presented additional cross-sectional analyses of the same sample at three [53], six [39,47,51], nine [39,47], and 12 [39] months post-tornado, each exploring a slightly different conceptual model. In addition, several other studies appeared to use overlapping samples [46,50,52,54]. Most of these studies presented mental disorder prevalence estimates among children who were exposed to the climate-related disaster but did not compare prevalence at different levels of exposure.

Among the remaining five studies, both Li et al. [48] and Peng et al. [37] investigated PTSD among children 7–15 years of age who were exposed to the 1998 flooding of Dongting Lake, an event that affected approximately 223 million people [59]. Both studies reported significant relationships between flood-related exposure severity and the presence of PTSD. For example, Li et al. [48] found that PTSD was significantly greater among children who experienced more severe flooding, were dropped or trapped in the water, witnessed somebody drown or die, were separated from family members, or had a relative or friend die (*p* ≤ 0.001). Quan et al. [44] and Zhen et al. [55] both investigated PTSD among a sample of 951 middle school students who were exposed to a rainstorm disaster at one week and two month follow-ups, respectively. One-week post-rainstorm, the prevalence of PTSD was 15.2% [44]. Further, both studies reported significant associations between rainstorm-related experiences of disruption/loss and PTSD (*p* ≤ 0.01) [44,55]. Finally, Wu and colleagues [42] studied 968 students who walked home during a snowstorm and reported a significant association between walking distance and PTSD (OR 1.01, 95% CI 1.00–1.01).

### 3.2. India

India was the location of three studies. Five years after a snowstorm and avalanche event, Chowhan and colleagues [40] interviewed 100 schoolchildren using the Mini International Neuropsychiatric Interview for Children and Adolescents. Fifty-four diagnoses were reported among 41 children, the most common being PTSD (14 cases), generalized anxiety disorder (GAD; 5 cases), separation anxiety disorder (4 cases), and major depressive disorder (MDD; 4 cases). Hassan et al. [43] conducted a mixed-methods study of 64 child survivors of flooding. In group discussions one month after the flood, the three most common themes were difficulty in concentration (discussed by *n* = 44 children), initial reactions to shock (*n* = 35), and flashbacks (*n* = 34). Neither of these studies compared outcomes between different levels of disaster exposure. Lastly, Kar et al. [41] studied PTSD and MDD among 447 schoolchildren who experienced a super-cyclone one year prior. Notably, children who lived in areas of higher exposure (i.e., areas submerged under water, with high levels of death and destruction) were significantly more likely to have PTSD compared to children in areas of lower exposure (OR 4.10, 95% CI 2.30–7.30).

### 3.3. Pakistan

Two studies were conducted in Pakistan. Ahmad et al. [38] captured flood-related PTSD among 522 randomly selected students. Four months after the event, more than 80% of the sample had PTSD according to a score ≥33 on the Impact of Events Scale—Revised [60]. In addition, students who were displaced by the flooding had significantly higher scores compared to those who were not displaced (*p* = 0.000). Sitwat and colleagues [36] used diagnostic interviews to identify the presence of mental disorders among 205 adolescent females approximately one year after a flood event. The prevalence of PTSD, GAD, and MDD was 2%, 1%, and 2%, respectively. All those who received a diagnosis had flood-related damage to their home.

### 3.4. Philippines

Two studies took place in the Philippines, both assessing the mental disorder effects of typhoon exposure. Mordeno and colleagues [56] measured acute stress disorder and depression among 225 child and adolescent survivors of Typhoon Washi, a severe tropical storm that led to extreme flooding and mudslides in the southern part of the country. The main goal of the study was to determine whether traumatic memories predicted negative mental disorder outcomes, and therefore the authors did not report mental disorder prevalence estimates. Nalipay and Moderno [57] measured PTSD among 446 Filipino college students three months after Typhoon Haiyan, which killed 6300 people [61]. Based on the PTSD Checklist for DSM-5 [62], 16% of the sample had PTSD.

### 3.5. Namibia

Using a cross-sectional design, Taukeni et al. [58] explored signs of PTSD among 429 schoolchildren two years after severe flooding in the Oshana region. PTSD was measured using the Child Trauma Screening Questionnaire. The main purpose of the study was to evaluate whether the post-disaster support services provided to schools were helping children cope, and therefore the authors did not compare prevalence of mental disorders at different levels of flood exposure.

### 3.6. Dominica

Lastly, Tavernier et al. [45] conducted a cross-sectional study of 174 local college students from Dominica six months after exposure to Tropical Storm Erika. Storm exposure was measured using a 7-item checklist adapted from the Hurricane-Related Traumatic Experiences scale and storm-related PTSD was measured using an 11-item checklist adapted from the PTSD Checklist. Notably, the bivariate analysis showed that PTSD was significantly correlated with greater tropical storm exposure (correlation coefficient 0.43, *p* < 0.05).

## 4. Discussion

Children living in LMICs are susceptible to the mental health burdens of climate change and climate-related disasters. Through our scoping review, we identified 23 studies exploring how exposure to climate-related disasters influences the presence of mental disorders among children in LMICs. Overall, there was a limited amount of evidence supporting an association between these two variables.

All studies included in this review investigated climate-related disaster exposures such as floods and typhoons, while none investigated broader climate change-related exposures such as gradual changes in temperature and precipitation. A similar pattern was identified in Sharpe and Davison [16] among an adult-focused LMIC population, where over 90% of the included studies explored climate-related disaster exposures. These exposures are often easier to quantify; climate-related disasters tend to have objective geographical boundaries and set start and end dates. Comparatively, indirect broader climate change exposures often occur over extended periods of time and place, making them more challenging to quantify. Future research should aim to examine the influence of the many diverse impacts of climate change on child mental health.

We found that few studies in our review tested the association between climate-related disasters and mental disorders. Although all included studies had a climate-related disaster exposure and mental disorder outcome, many tested complex conceptual models that did not directly explore our association of interest (e.g., [47,50,56]). However, we chose to include these studies due to their relevance to the topic area; there is a dearth of child-specific literature focusing on these particular variables. Further, none of the studies compared mental disorders between groups who were exposed to a climate-related disaster and groups who were unexposed, making it challenging to understand the true association between these two variables. Existing reviews have also identified a lack of evidence on the association between climate change and child health in LMICs [28,63]. Ultimately, more well-designed studies are needed, especially those making unexposed group comparisons that allow for direct tests of the association between climate change or climate-related disasters and mental health. This being said, it is also important to acknowledge that many of the studies identified by our review took place in post-disaster and under-resourced settings, posing various challenges such as physical inaccessibility and loss of records [64]. Despite these barriers, other studies have been successful in comparing mental disorders between those exposed and unexposed to a climate-related disaster event. For example, in an adult population, Amstadter and colleagues [65] collected mental health data before and after the occurrence of Typhoon Xangsane in the Philippines, allowing for a pre–post comparison. Wind and colleagues [66] compared anxiety and depression scores between adults who were exposed to a flood event and a similar group who was not exposed. These examples provide potential study designs to be applied in child populations to gain a better understanding of the relationship between climate-related disasters and mental disorders.

Despite the lack of unexposed group comparisons, several studies assessed climate-related disaster exposure severity (i.e., level of exposure) as a risk factor for mental disorders, generally identifying significant exposure-outcome associations [37,38,41,42,44,45,48,49,55]. A narrative review conducted by Burke et al. [25] highlighted similar findings, particularly among children living in LMICs. A number of reviews have also recognized similar trends among adult populations. For example, Rataj et al. [14] identified increased burdens of PTSD, anxiety, and depression associated with climate-related disasters in developing countries. Future research efforts should focus on expanding this evidence base for children specifically given that the biological pathways through which climate change affects mental health may be distinct among this group, potentially leading to lasting or compounded outcomes throughout the life course [20,21,22,23]. Although our findings provide some evidence supporting a relationship between climate-related disasters and increased mental disorders among children in LMICs, further child-specific information is needed.

The studies included in our review used a variety of measures to assess mental disorder outcomes. While some studies conducted formal assessments using diagnostic interviews, such as the Mini International Neuropsychiatric Interview for children and adolescents [40], others used informal screening tools, such as the Childhood Trauma Screening Questionnaire [58] or the Depression Self-Rating Scale for Children [56]. Although the validity and reliability of screening tools may vary, they play an important role in rapidly identifying mental health problems among children in post-disaster settings [67]. Further, screening tools are particularly useful in LMIC settings, where health care workers may not have the qualifications or the time to conduct formal diagnostic interviews [68].

In addition, 21 of the 23 studies included in this review used a cross-sectional design. Cross-sectional designs may lack temporality, where in some contexts it becomes challenging to identify whether the climate-related disaster exposure truly preceded the mental disorder outcome. Included in our review, An et al. [39] and An et al. [47] conducted prospective cohort studies that measured PTSD among a group of middle school students at 6, 9, and 12 months after a tornado. Longitudinal approaches such as these may help to identify the changing effects of climate-related disasters on mental disorders over time, providing important evidence for optimizing the timing and duration of post-disaster childhood mental health interventions.

The patterns identified in this review largely aligned with those identified by a similar review focused on an adult LMIC population [16]. More specifically, both reviews identified some limited support for the relationship between climate-related disasters and increased mental disorders, although the evidence base is considerably smaller among children. While more research is needed to understand the child-specific impacts of the changing climate, various adaptation policies and mental health interventions are currently being researched and implemented. In disaster contexts, there is child-specific evidence supporting the effectiveness of cognitive behavioral therapy, narrative exposure, meditation relaxation, group therapy, and play therapy for improving mental health [67,69]. Kar et al. [67] also emphasized the need to train disaster relief workers to appropriately support children in these situations, for example through the use of emotional first aid. In the more general context of climate change, UNICEF [17] recommends investments in childhood education, health, nutrition, and poverty reduction. For example, they predict that investments in education, through creating infrastructure resilient to disasters and expanding schooling so that children are empowered to mitigate climate change, could reduce climate risk for more than 275 million children [17]. These actions will also subsequently reduce potential negative mental health impacts.

### Strengths and Limitations

Our review had several strengths. First, we performed comprehensive searches of both the grey and academic literature. In conjunction with a health sciences librarian, we developed a detailed search strategy to capture literature from five relevant electronic databases (MEDLINE, Embase, APA PsycInfo, Global Health, and Sociological Abstracts). We also searched recent issues of relevant journals and performed reference snowballing for all included studies. We used a three-step strategy to search the grey literature, although it did not return any results. Similarly, Charlson et al. [6] performed a systematic search of climate change and mental health among all populations and also did not identify any relevant grey literature, suggesting that our search was rigorous. Second, our work was novel given that no previous reviews have explored this specific topic. While others have investigated climate change, climate-related disasters, and mental health among individuals in LMICs [14,16,70], none have explored this topic specifically among children. Children in LMICs are susceptible to the negative mental health impacts of climate change, thus we felt it was important to capture the current state of this literature to identify gaps and determine important next steps for research.

This review also had limitations. For one, we only included studies written in the English language. This decision was made due to project feasibility; however, non-English studies making useful contributions to the research question were therefore excluded. Our search was also limited to studies published in 2007 or later. We made this decision because the Intergovernmental Panel on Climate Change had not acknowledged the effects of climate change on human health and well-being before this date. Thus, we would have missed any relevant studies published before 2007. However, Charlson et al. [6]’s systematic search was performed from 2001 onwards and did not identify any studies published before 2007, suggesting that we captured the full extent of the literature. Given the close relationship between climate change and extreme weather events [2], we created an evidence-based list of climate-related natural disasters to include in this review (Appendix A). While it is reasonably likely that climate change played a role in many of these disasters, some may have occurred for unrelated reasons. This decision may have resulted in several false-positive studies; however, we felt this was justified given that the effects of climate change are challenging to define and measure. In addition, as our search focused on mental disorders, it did not explore all concepts related to mental health or well-being. Suldo and Shaffer [71] and Antaramian et al. [72] present the dual-factor model of mental health, which suggests that while mental health has often been described in relation to the presence or absence of psychopathology, it should also be represented by positive indicators such as well-being. For feasibility reasons, we limited our search to only include mental disorder outcomes. Future research should include subclinical mental illness, mental health, well-being, and resilience outcomes among children in LMICs. Lastly, we did not formally assess study quality using a risk of bias tool. Formal quality assessments are not typically performed in a scoping review [31]. Informally, we observed substantial variation in study quality. Some studies provided little to no information on sampling techniques, measurement tools, and/or statistical analysis. Nonetheless, the main objective of our review was to capture the scope of the literature, regardless of its quality.

## 5. Conclusions

Climate change and its impacts represent a substantial threat to children, especially those living in LMICs. We used the scoping review methodology to determine how exposure to climate change and climate-related disasters influences the presence of mental disorders within this population. We found that the literature in this area was fairly sparse. Importantly, more well-designed studies on this topic are required, including those with unexposed comparison groups that avoid issues related to temporality. Improvements in the quality and quantity of research will create an evidence base to support policy and interventional efforts for reducing the negative mental health effects of climate change.

## Figures and Tables

**Figure 1 ijerph-19-02896-f001:**
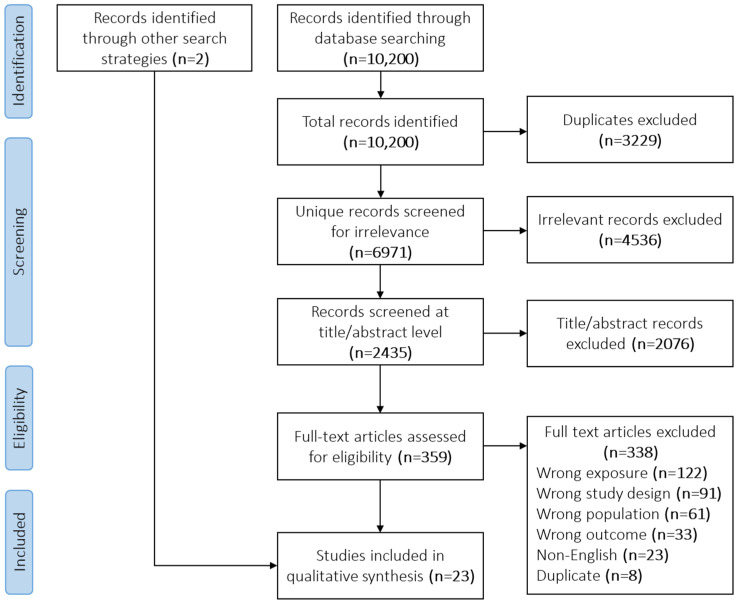
PRISMA diagram showing the study inclusion process.

**Figure 2 ijerph-19-02896-f002:**
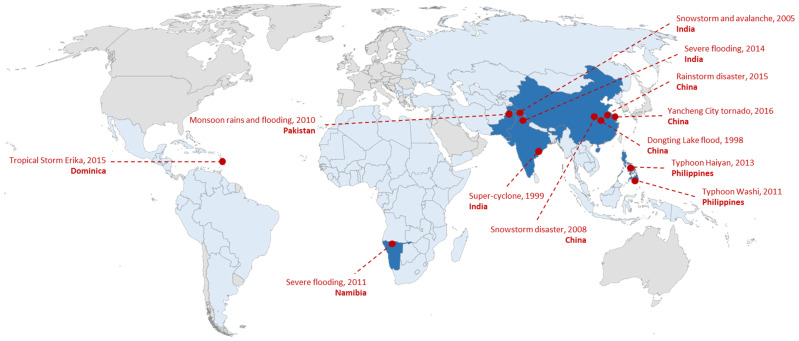
World map showing the 12 unique climate change exposures captured by the included studies (dark blue countries). The climate change events took place in China (*n* = 4), India (*n* = 3), Philippines (*n* = 2), Pakistan (*n* = 1), Namibia (*n* = 1), and Dominica (*n* = 1). The remaining low- and middle-income countries are colored light blue and high-income countries are colored grey.

**Figure 3 ijerph-19-02896-f003:**
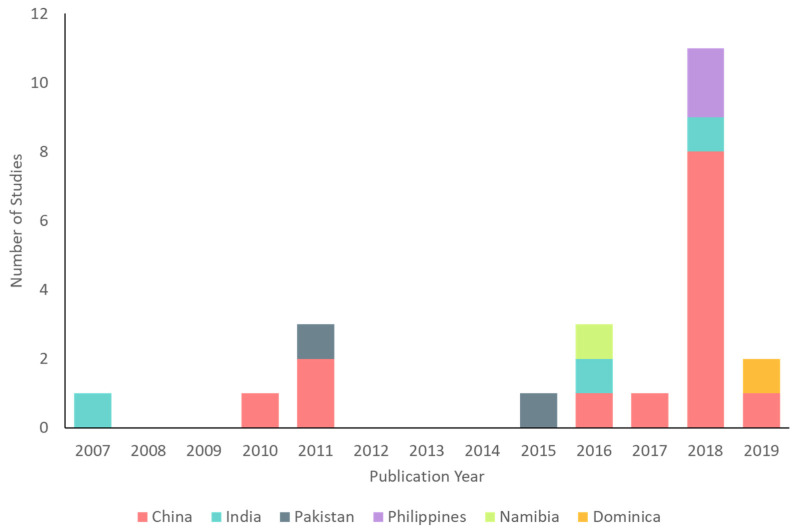
Included studies (*n* = 23) by year and country.

**Table 1 ijerph-19-02896-t001:** Scoping review inclusion and exclusion criteria.

	Inclusion Criteria	Exclusion Criteria
Population	- Child focused (study population with mean age 18 years and under) - Study population located entirely in low- and middle-income countries at time of exposure (may be located in more than one) ^1^	- Adult focused (study population with mean age over 18 years) - Any part of the study population located in high-income countries
Intervention (Exposure)	- Climate-related disaster exposure (see Appendix A for list of climate-related disasters) OR climate change-related exposure (as identified by the study author) ^2^	
Comparison	- Any	
Outcome	- Mental disorders (anxiety, depression, post-traumatic stress, acute stress, substance use and addiction, bipolar, schizophrenia, suicidal behavior, non-suicidal self-injury) evaluated based on DSM or ICD symptoms	- Mental disorders not evaluated based on DSM or ICD symptoms - Study only measures positive mental health
Study Design	Any type of empirical literature, including - Journal articles (quantitative, qualitative, or mixed methods) - Grey literature (conference proceedings, dissertations, government and organization documents, policy briefs)	- Narrative reviews, syntheses (scoping reviews, systematic reviews, meta-analyses, etc.), commentaries, editorials, expert opinions - Validation studies - Non-English language - Date of publication before 2007 ^3^

Abbreviations: Diagnostic and Statistical Manual of Mental Disorders (DSM); International Classification of Diseases (ICD). ^1^ Based on the World Bank’s designation of a low- and middle-income country (includes low-, lower-middle, and upper-middle-income economies) [33]. ^2^ Climate change is a challenging concept to define and measure. We chose to include studies where the study author explicitly mentioned that they were investigating the effects of climate change (e.g., used the words “climate change” when describing the study exposure). ^3^ The Intergovernmental Panel on Climate Change’s fourth assessment report (AR4) was published in 2007, in which the authors began to acknowledge the impacts of climate change on human health and well-being. This approach was taken in a similar scoping review by Middleton and colleagues [34].

**Table 2 ijerph-19-02896-t002:** Summary of all included studies (*n* = 23). All studies were quantitative unless otherwise specified.

Study ID	Design	Sample Size and Population	Sampling Method	Climate-Related Disaster Exposure(s)	Mental Disorder Outcome(s)	Outcome Measurement Tool(s)	Post-Exposure Follow-Up Length	Main Findings (on the Association between Climate-Related Disasters and Mental Disorders)	Mental Disorder Prevalence Estimates
China (*n* = 14)									
An-2019 (same baseline sample as Xu 2018a) [39]	Prospective cohort	154 middle school students	Multistage cluster random sampling	Tornado	PTSD, depression	CPSS, CES-DC (self-reported surveys)	6 (T1), 9 (T2), and 12 (T3) months		Post-tornado PTSD and depression: 55.84% and 56.49% at T1, 50.0% and 65.58% at T2, 47.40% and 66.01% at T3; PTSD at T1 significantly predicted depression at T2 (*p* < 0.001) and PTSD at T2 significantly predicted depression at T3
An-2018a (sample related to Xu-2018a) [46]	Cross-sectional	443 junior high school students	Multistage cluster random sampling	Tornado	PTSD	CPSS (self-reported survey)	12 months		
An-2018b (same baseline sample as Xu 2018a) [47]	Prospective cohort	204 middle school students	Multistage cluster random sampling	Tornado	PTSD	CPSS (self-reported survey)	6 and 9 months		
Li-2010 [48]	Cross-sectional	4327 children aged 7–15 and their parents	Multistage cluster random sampling	Flood (Dongting Lake)	PTSD	DSM-IV criteria (interview)	18 months	Presence of PTSD was significantly greater among children who experienced flash or drainage problem flooding, experienced moderate (25–49% of total village area) or severe (≥50% of total village area) flooding, were dropped into water, were trapped in water, had a serious injury, had seriously injured relatives, witnessed somebody drown, had death of a family member or friend, were trapped in water near a dead body, had previous flood experience, were separated from family members, had teachers or classmates drown, had class suspended, had the following school semester postponed, and had parents with PTSD (all *p* ≤ 0.001); multivariate analyses showed that having a PTSD positive father (OR 3.0, 95% CI 2.05–4.50) or mother (OR 4.4, 95% CI 2.99–6.45) significantly increased risk of child PTSD	Post-flood PTSD: children 4.7%, parents 11.2%
Peng-2011 [37]	Cross-sectional	7038 children aged 7–15	Multistage cluster random sampling	Flood (Dongting Lake)	PTSD	DSM-IV criteria (interview)	~18 months	Flood type (flash > collapsed > soaked) and whether school reopening was delayed (yes > no) were significantly associated with PTSD (*p* < 0.001)	Post-flood PTSD: 2.05%
Quan-2017 [44]	Cross-sectional	951 middle school students	Multistage cluster random sampling	Rainstorms	PTSD	PCL-5 (self-reported survey)	1 week	Presence of PTSD was significantly correlated with rainstorm-related experiences and perceived severity of disaster (*p* = 0.01)	Post-rainstorm PTSD: 15.2% identified as probable cases
Wu-2011 [42]	Cross-sectional	968 students who walked home during storm	Convenience sampling	Snowstorm	PTSD	IES-R (self-reported survey)	3 months	Walk time (5+ hours > 2–5 h > 0–2 h) and walk distance (20+ km > 10–20 km > 0–10 km) were significantly associated with PTSD (*p* < 0.01); binary analyses showed walk distance (OR 1.01, 95% CI 1.00–1.01) significantly increased odds of PTSD	Post-snowstorm PTSD: 14.5%
Xu-2018a [49]	Cross-sectional	247 middle school students (grades 7–9)	Multistage cluster random sampling	Tornado	PTSD, depression	CPSS, CES-DC (self-reported surveys)	3 months	Significantly greater odds of PTSD among children who had injured relatives/friends (OR 1.98, 95% CI 1.01–3.92), feared injury/death (OR 1.92, 95% CI 1.14–3.24); significantly greater odds of depression among children who had injured relatives/friends (OR 2.05, 95% CI 1.03–4.07)	Post-tornado PTSD: 57.5%, depression: 58.7%
Xu-2018b (sample related to Xu-2018a) [50]	Cross-sectional	431 middle school students	Multistage cluster random sampling	Tornado	Depression	CES-DC (self-reported survey)	9 months		
Xu-2018c (same sample as Xu-2018a) [51]	Cross-sectional	247 middle school students	Multistage cluster random sampling	Tornado	PTSD, depression	CPSS, CES-DC (self-reported surveys)	6 months		
Yuan-2018a (sample related to Xu-2018a) [52]	Cross-sectional	431 middle school students	Multistage cluster random sampling	Tornado	PTSD	CPSS (self-reported survey)	9 months		
Yuan-2018b (same sample as Xu-2018a) [53]	Cross-sectional	247 middle school students	Multistage cluster random sampling	Tornado	PTSD	CPSS (self-reported survey)	3 months		
Zhang-2018 (sample related to Xu-2018a) [54]	Cross-sectional	443 middle school students	Multistage cluster random sampling	Tornado	PTSD	CPSS (self-reported survey)	12 months		
Zhen-2016 (same sample as Quan-2017) [55]	Cross-sectional	951 middle school students	Multistage cluster random sampling	Rainstorms	PTSD	PCL-5 (self-reported survey)	2 months	PTSD was significantly correlated with severity of rainstorm-related experiences (*p* < 0.001)	
India (*n* = 3)									
Chowhan-2016 [40]	Cross-sectional	100 children aged 6–17 from local school	Systematic sampling	Snowstorm, avalanche	DSM-IV disorders	MINI-KID (interview)	5 years		Observed 54 post-event diagnoses among 41 patients: PTSD (14), GAD (5), separation anxiety disorder (4), MDD (4), dysthymia (3), agoraphobia (3), social phobia (3), adjustment disorder (3), suicidality (2), PD (2), mania (1), specific phobia (1), substance abuse (1)
Hassan-2018 (mixed methods) [43]	Cross-sectional	64 children who had resumed schooling	Convenience sampling	Flood	PTSD	Quantitative: CRIES-8; Qualitative: group discussions (interviews)	1 month		Main qualitative themes: initial reactions to shock, intrusion, flashbacks, avoidance, difficulty in concentration, and helplessness and sadness
Kar-2007 [41]	Cross-sectional	447 students	Multistage stratified cluster random sampling	Cyclone	PTSD, MDD	Psychiatrist evaluation using ICD-10-DCR criteria (interview)	12 months	Exposure level was significantly associated with PTSD (high vs. low OR 4.10, 95% CI 2.30–7.30)	Post-cyclone PTSD: 30.6% (an additional 13.6% considered subsyndromal); MDD: 23.7% (comorbid with PTSD in 34.3% of this group)
Pakistan (*n* = 2)									
Ahmad-2011 [38]	Cross-sectional	522 students aged 10–16	Random sampling	Flood	PTSD	IES-R (self-reported survey)	4 months	PTSD score significantly higher among those who were displaced vs. those who were not displaced (*p* = 0.000)	Post-flood PTSD: 3.06% none, 14.17% partial, 8.81% probable, 73.94% high
Sitwat-2015 [36]	Cross-sectional	205 females aged 13–19	Purposive sampling	Flood	PTSD, GAD, MDD	Diagnostic interview using DSM-IV-TR (interview)	~12 months		Post-flood PTSD: 2%, GAD: 1%, MDD: 2%
Philippines (*n* = 2)									
Mordeno-2018 [56]	Cross-sectional	225 child and adolescent survivors at evacuation centers	Convenience sampling	Typhoon (Washi)	ASD, depression	ASDI (interview), DSRS-C (self-reported survey)	1 month		
Nalipay-2018 [57]	Cross-sectional	446 college students	Convenience sampling	Typhoon (Haiyan)	PTSD	PCL-5 (self-reported survey)	3 months		Post-typhoon PTSD: 16.14%
Namibia (*n* = 1)									
Taukeni-2016 [58]	Cross-sectional	429 students aged 8–18	Stratified sampling	Flood	PTSD	CTSQ (self-reported survey)	2 years		Post-flood PTSD: 72.8% of children 13+, 55.2% of children <13
Dominica (*n* = 1)									
Tavernier-2019 [45]	Cross-sectional	174 college students	Not specified (sample from a larger study)	Tropical storm (Erika)	PTSD	Adapted PTSD Checklist (self-reported survey)	6 months	PTSD was significantly correlated with severity of tropical storm exposure (*p* < 0.05)	

Abbreviations: ASD (acute stress disorder), ASDI (Acute Stress Disorder Interview), DSM (Diagnostic and Statistical Manual of Mental Disorders), DSRS-C (Depression Self-Rating Scale for Children), CES-DC (Center for Epidemiologic Studies Depression Scale for Children), CPSS (Child PTSD Symptom Scale), CRIES-8 (Children’s Revised Impact of Event Scale), CTSQ (Child Trauma Screening Questionnaire), GAD (generalized anxiety disorder), ICD-DCR (International Classification of Diseases—Diagnostic Classification for Research), IES-R (Impact of Events Scale—Revised), MDD (major depressive disorder), MINI-KID (Mini International Neuropsychiatric Interview for children and adolescents), PCL-5 (PTSD Checklist for DSM-5), PD (panic disorder), and PTSD (post-traumatic stress disorder).

## Data Availability

The data presented in this study are available within this article and its Appendix A.

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
