# Peer review of "A Scoping Review of Climate Change, Climate-Related Disasters, and Mental Disorders among Children in Low- and Middle-Income Countries"

_ijerph, 2022, doi:10.3390/ijerph19052896_

Round 1
Reviewer 1 Report
The authors have prepared a scoping review of relevant literature currently available on the topic of climate change/disasters and subsequent mental illness in children. They have previously published a similar review on the topic in adults, and this paper extends this previous information and is also novel because there has not previously been a review of the current literature on this specific topic. The authors clearly describe the methodology used to identify the available literature. They provide concise summaries of the papers reviewed, and a discussion of the current limitations of the literature and suggestions for future studies needed to better understand whether climate change will contribute to an increase in mental illness among children, specifically in low and middle income countries. This is an important topic as I agree with this authors that more work is needed to study this issue in order to identify strategies to address the impacts of climate change. I recommend publication of this review.
Author Response
Thank you very much for your positive and thorough review. We have made a number of edits in response to other reviewer comments.
Reviewer 2 Report
A scoping review of climate change, climate-related disasters, and mental disorders among children in low- and middle-income countries
The paper reviews literature about climate change related natural disasters and its impact on the mental health of children (or does it pertain young adults?) in Low and Middle income countries. They assume that the impact in of natural disaster is these countries is more severe and that the impact is even more severe for children (but why is that the case?? What underlying variable causing such a relationship?). The aim of the study is implicit (expect for doing the review) and therewith it is unclear what focus the review takes.
In my opinion, the research is very relevant, since the effect of climate change related natural disasters on the mental health is not often a focus of study, but in my opinion a very relevant one, since coping with natural disasters asks for resilience of communities, and this is often not that high in low or middle income countries (regions). Moreover traumatic events can result in Post traumatic stress, that deteriorates the resilience even more. Since children rely more on the resilience of their community, they are more susceptible to long lasting impacts of the disaster (this can be material, physical as well as mental).
In my opinion the aim of the research should be stated in academic terms of reviewing the literature in order to come up with a preliminary framework to study the impact of natural disaster on (community) resilience as well as mental health issues of children involved. s including mental health). By stating this aim, criteria to review the selected papers are: academic quality of the research, definition, conceptualization and operationalization of the key variables in the framework (of the key variables in the study related to the framework) and the findings (the relations between the key variables of the framework).
Main comments:
- Defining the research problem:
- Why are LMIC countries of special interest, is it about the facilities provided (that is something external) or is it because of the lower community resilience (a state of mind of a community; the capability to recover from a disaster. That depends partly on the facilities but is mainly a state of mind or refers to the social fabric of a community)
- By assuming that external events (facilities, natural disasters) impact people, one makes the implicit assumption that the impact of a specific event is similar of all people involved, but that is not the case and it does not depend on demographic variables, but on the state of mind of the person, social fabric of the community.
- The key constructs ware weakly defined: so when is a natural disaster a climate change related disaster. What is mental health and why is the focus on mental disorders interesting given the research objective?
- Materials and methods.
- The operationalization of the selection criteria for the papers deviate from the section criteria in the research question: average age of 18 of a sample, does not refer to children. Why are those specific mental disorders selected. Some refer to states of mind, others to actions. What is the inclusion criteria for low and middle income countries (when is a country considered low/middle income, moreover in my opinion that depends also on the region)
- No inclusion criteria refer to the scientific quality of the research: type of research (descriptive, analytical), sampling (size as well as sampling procedure), operationalization of the key variables (and are those comparable across studies??), statistical testing used (and is it appropriate given the distribution of the data in the research).
- The screening of the papers is unclear: what are the search terms used? What are the exact criteria (so their operational definition) used in the successive screenings.
- results
- In my opinion the result section could benefit is for each of the 23 studies the methodological characteristics are indicated: a/ measurements in time [post test only (+ time interval), retrospective study, pre test, post test design; time series (after the disaster), b/ data collection (interview, survey), c/ sampling (kind of sampling procedure, and sample size). d/ definition and operationalisations used for mental health and other key variables of the study that could be of importance for the conceptual framework (such as severeness of the natural disaster, facilities, resources of the community, community resilience), e/ kind of analyses (what statistical procedure). Table 2 is a good start but some information is missing.
- If the aim of the review is to come up with a conceptual framework, it is odd that the results are discussed by location. In a conceptual framework, one is mainly interested in general knowledge that is applicable to all contexts to which the framework pertains (natural disasters in low/middle income countries). So I would suggest to structure the results section along the key variables identified in the different studies that provide the building blocks for the conceptual framework.
- Discussion/conclusion
- In the discussion limitations of the reviewed study are discussed. In my opinion these limitations should be discussed in the ‘results’ section. Since it are not limitations of the review study itself, but of the studies reviewed. In the discussion, I expect a conceptual framework that includes the main variables attended to in the reviewed studies, their definition, conceptualization and operationalization, and the expected relations between these variables. These relations are only preliminary since most studies do not use a baseline.
- In my opinion it is far fetched to suggest various adaptation policies (which can be regarded as an intervention) if not conceptual framework exist how and when natural disasters affect mental health of children in low/middle income countries. The conceptual framework can be a starting point of feasibility studies to design, monitor and adjust policies that could be effective in this regard.
- Based on the literature review (as it is presented) no claims can be made that climate change affects the mental health of children (line 408-409)
- More detail comments and suggestions are added as comments in the paper.

Author Response
Thank you for your thorough review. We have constructed the following table with your comments and our responses. We also attach a copy of the manuscript with our responses to your notes that were included directly on the previous draft. Please let us know if you have any remaining concerns. Thank you again, your feedback has helped us strengthen the manuscript.
|
Reviewer #2 |
|
|
We would like to thank the reviewers for their comment. We have added some clarification to our rationale for focusing on low- and middle-income countries, namely that these countries are susceptible to the negative effects of climate change and may have limited resources and ability for mitigation and adaptation (lines 46-52).
We agree with the reviewers that resilience is also a key consideration when exploring the effects of climate change and climate-related disasters on a community. To address this, we have added 1) acknowledgment of the importance of individual and community-level resilience, and 2) justification of why the broader context may place specific populations at potentially greater risks for negative mental health impacts (lines 52-58). |
|
|
Please see our response to the above comment where we have addressed the role of resilience. |
|
|
Thank you for your comment. We believe that the review’s constructs are adequately defined in the methods section. Specifically, the eligibility criteria are outlined in Section 2.1 and Table 1.
As mentioned in Table 1, the list of climate-related disasters can be found in Table S1 (supplementary material). We have now bolded this statement to make it more clear. Table S1 lists which disasters we considered as climate-related and provides a brief rationale for this structure.
Lines 86-90 present a rationale for focusing on mental disorders. We chose to focus on mental disorders because this is a particularly burdensome aspect of mental health. We acknowledge in our limitations section that mental disorders only represent just one aspect under the broader construct of mental health, stating that future research should explore other aspects of this construct, including resilience (lines 429-436) |
|
|
In terms of the age group cut-off, we realize that an average age is not typically used. In conjunction with our previously published review on adult-focused populations (doi: 10.1136/bmjopen-2021-051908), we found it was important to capture studies that primarily included children (<18 years) but also may have included a small proportion of adults (≥18 years), given that this situation was common within our area of literature. This rationale has been added to the eligibility criteria section (lines 108-112).
We selected our mental disorders of interest upon reviewing all disorder types listed in the DSM-5. We used our prior knowledge of this field to determine which disorders may be important in the context of climate change or climate-related disasters (i.e., whether there had been previous research on a certain disorder in other populations or a biological rationale behind it). The list of included disorders is presented in lines 114-116 and in Table 1. As suggested by the reviewer in the PDF comments, we have added a reference to the DSM in line 116.
In terms of the inclusion criteria for LMICs, please refer to the first footnote in Table 1 (lines 123-124). We classified countries based on the World Bank’s designation of LMICs. We agree with the reviewer that this may represent a somewhat arbitrary form of classification; however, we chose this approach due to its objectivity and standard use in the literature. |
|
|
We appreciate this suggestion. To our knowledge, ours was the first review to explore the extent of the literature on this specific topic. Therefore, our main objective was to capture and present the full extent of the literature in this area, instead of focusing solely on the high-quality research. As our results showed, currently there is very little research on this topic. Therefore, restricting the eligibility criteria further would leave us with few or no included studies.
In relation to this, we have added a point to the limitations section explaining how we chose not to formally assess the quality of our included studies, although many were likely of poor quality (lines 436-441). Ideally, at a time when the literature is more developed, future reviews should focus on high-quality studies, using guidelines similar to those mentioned by the reviewer here. |
|
|
All search terms are presented in the supplementary material (Tables S2-S9). Section 2.2 provides a detailed outline of the search strategy and Section 2.3 discusses the process used to screen and select the studies. Studies were included based on the eligibility criteria presented in Table 1, which we have now clarified in the description of the screening process (lines 164, 166). |
|
|
We thank the reviewer for their suggestion. a) This information is available in Table 2. Under the ‘Design’ column, we list the study design for each included study (cross-sectional, prospective cohort, etc.). Under the ‘Follow-Up Length’ column, we list the measurements in time for each study. To clarify this, we have changed the title of the column to ‘Post-Exposure Follow-Up Length’. b) We have specified the data collection method under the ‘Outcome Measurement Tool(s)’ column in Table 2. Data were collected either via an interview or a self-reported survey. c) The sample sizes can be found under the ‘Sample Size and Population’ column in Table 2. We have added a column in Table 2 (called ‘Sampling Method’) that describes the sampling method of each study. d) In Table 2 under the ‘Mental Disorder Outcome(s)’ and ‘Outcome Measurement Tool(s)’ columns, we list the mental disorder constructs measured and the specific tools used to measure them. In Table 2 under the ’Climate-Related Disaster Exposure(s)’ column we list the type of disaster and in Figure 2 we present these disasters on a world map. With regards to the additional variables for informing a conceptual model, we believe this is beyond the scope of our project. For more details, please see our response to the following comment. e) We chose not to include this information, given that it was not consistently reported by the included studies. We have now addressed this in the limitations section, where we have added a point explaining how we chose not to formally assess the quality of our included studies, although many were likely of poor quality (lines 436-441). |
|
|
As our review showed, the literature on this particular topic is quite sparse and heterogeneous. At this stage, we do not believe that there is adequate evidence to develop a proper conceptual framework. Our review’s objective was to present the literature on this topic, identifying existing gaps and areas for future investigation. Therefore, we believe that the development of a conceptual framework is beyond the scope of this project. As the literature becomes further developed in the coming years and decades, future reviews should aim to create such a framework. |
|
|
As stated in lines 92-93, one of our objectives was to identify and summarize existing gaps in this area of literature. As such, we felt that it is appropriate to use the discussion to summarize and describe the gaps of this area of literature. The limitations section, lines 415-441, presents limitations related to the review process itself.
Please see the above point regarding the conceptual framework. |
|
|
As mentioned in our previous responses, we believe that a conceptual framework is beyond the scope of this project given the current state of the literature. We agree that a conceptual framework would be a useful tool for informing interventions and adaptation policies and should be pursued in future.
In lines 386-388, we describe that “While more research is needed to understand the child-specific impacts of the changing climate, various adaptation policies and mental health interventions are currently being researched and implemented.” We go on to present existing interventions and policies, without linking them specifically to the findings of our review. In the conclusion, we reiterated that additional research is needed to create a stronger evidence base, which then can be used to support policy and interventional efforts (lines 448-450). In line 84, we added the words “may eventually” to indicate that interventions and adaptations may be informed at a future date. |
|
|
Thank you for bringing this to our attention. We have removed the word “particularly” from the sentence, so that it now reads “Children in LMICs are susceptible to the negative mental health impacts of climate change, thus we felt it was important to capture the current state of this literature to identify gaps and determine important next steps for research" (line 412). All people are susceptible to the negative mental health impacts of climate change, and we were interested in capturing the research on this specific population given that others have yet to do so.
In line with this, we have made similar edits to line 310, removing the word ‘extremely’. We have also made similar edits in the introduction section, removing the word ‘particularly’ from line 45, removing the word ‘disproportionately’ from line 62, and changing ‘are’ to ‘can be’ in line 63. In addition, we removed the following sentence from line 70: “Children living in LMICs are particularly susceptible to the burdens of climate change and climate-related disasters”. |
|
|
We thank the reviewers for providing these detailed comments on our manuscript. Please see the attached file for our point-by-point responses to each of the 51 comments. Please note that the line numbers mentioned in these comments refer to the revised draft of the manuscript. |
|

Reviewer 3 Report
Dear Authors
This manuscript “A scoping review of climate change, climate-related disasters, and mental disorders among children in low- and middle-income countries” (ijerph-1544533) joins two highly relevant aspect of Public Health and Epidemiology: infant mental health and climate change or extreme meteorological events (related or not with climate change).
In my opinion, there are key methodological questions that this manuscript has not considered and could consequently; the conclusion could be strongly biased from reality.
First, in the title of your manuscript you talk about ·”Climate change” but, as you say, you do not find any article talking about this topic. So, why do you include the term “climate change” in the title and in the aim of the study”. Second aspect, consequence of the first one: How do you differentiate “Extreme climate events” from “Extreme meteorological events”?. Extreme meteorological events occur every year but to be considered “climatic” you should have a more comprehensive approach or be defined on an operational or theoretical framework or based on definitions of an international organism or institution or,… . Thus, I would say that you talk about the effects of extreme meteorological events on mental health in LMICs. The consequence would be that this study represents a different theoretical framework.
You define among your criteria for a study to be included in the analysis of the manuscripts that it must be written in English. Here my point of view is the following: 1) If you think that many of these events happen in low middle income countries, 2) if it is highly probably that these countries publish their studies in other languages , 3) if you know that there is scarce literature on this topic: don’t you think that perhaps in this situation, not considering other languages is detrimental for your review?. Using English for a review could be highly reasonable if the literature of the issue is abundant. Many and many reviewers use other languages because they know them or because they translate them; this has happen with many articles written, for example, in Chinese, French. Spanish, German, … . I strongly recommend using other languages for your aim.
You comment that you did not find grey literature. My comment is the same that in the previous paragraph. Probably most of it, if really there is, would be found in other languages.
In the main table, I am not sure if the articles are independent, many based in the same event and with the same or similar sample, (different follow-ups should be summarized). Many studies do not show data of mental health. So, are some these studies different or the same or very similar studies?.
To end, my recommendation would be to take into consideration as much studies as possible (different languages), to include relevant information of the event, magnitude and impact of the event and to include only studies with information of the event and of the impact in mental health.
Author Response
Thank you very much for the thorough review. We have constructed the following table with your comments and our responses. Please let us know if you have any remaining concerns.
|
Reviewer #3 |
|
|
First, in the title of your manuscript you talk about ·”Climate change” but, as you say, you do not find any article talking about this topic. So, why do you include the term “climate change” in the title and in the aim of the study”. |
Thank you for pointing this out. While we did not find any studies with a climate change-specific exposure, we did include this concept in our a priori objectives and search strategy. Therefore, we believe that it is equally as important to acknowledge the lack of studies in this area as it would be to present the results of studies if we had found any. In lines 315-324, we hypothesize why this may have occurred.
In conclusion, we believe that it is important to acknowledge this specific gap in the literature. |
|
Second aspect, consequence of the first one: How do you differentiate “Extreme climate events” from “Extreme meteorological events”?. Extreme meteorological events occur every year but to be considered “climatic” you should have a more comprehensive approach or be defined on an operational or theoretical framework or based on definitions of an international organism or institution or,… . Thus, I would say that you talk about the effects of extreme meteorological events on mental health in LMICs. The consequence would be that this study represents a different theoretical framework. |
We thank the reviewer for pointing out this distinction. We agree that it can often be challenging to determine the true causal role that climate change may play in these weather events. In Table S1 (supplementary material), we classify weather events as ‘climate-related’ or ‘not climate-related’. This classification was based on previous categorizations in the literature. We have added references to the sources that helped inform our classifications. We also added a mention of this issue to our limitations section, describing that while it is reasonably likely that climate change played a role in many of these disasters, some may have occurred for unrelated reasons (lines 423-429). |
|
You define among your criteria for a study to be included in the analysis of the manuscripts that it must be written in English. Here my point of view is the following: 1) If you think that many of these events happen in low middle income countries, 2) if it is highly probably that these countries publish their studies in other languages , 3) if you know that there is scarce literature on this topic: don’t you think that perhaps in this situation, not considering other languages is detrimental for your review?. Using English for a review could be highly reasonable if the literature of the issue is abundant. Many and many reviewers use other languages because they know them or because they translate them; this has happen with many articles written, for example, in Chinese, French. Spanish, German, … . I strongly recommend using other languages for your aim. |
Thank you. We agree that our search may have missed some important studies that were not written in English. As mentioned in the methods section, we chose to limit our search to English-only due to feasibility constraints related to this project (lines 118-119). Future reviews should aim to include studies in multiple languages.
To further highlight this drawback to our research, we have included it as a limitation in the discussion section (lines 415-417). |
|
You comment that you did not find grey literature. My comment is the same that in the previous paragraph. Probably most of it, if really there is, would be found in other languages. |
Please see response to above comment. |
|
In the main table, I am not sure if the articles are independent, many based in the same event and with the same or similar sample, (different follow-ups should be summarized). Many studies do not show data of mental health. So, are some these studies different or the same or very similar studies?. |
Thank you for pointing this out. This is correct; several studies used the same or overlapping samples. We indicated these cases in the first column of Table 2 (for example, in the first study An-2019, we indicated that they used the same baseline sample as Xu 2018a). To make this more clear, we have bolded these instances throughout the table.
Several studies did not explicitly show data on mental health. Although these studies measured a mental disorder outcome (and therefore were eligible to be included in our review), they did not present this data in a way that was useful to us. Instead, they focused on other objectives. For example, Mordeno and colleagues (reference #56) measured acute stress disorder and depression among 225 child and adolescent survivors of Typhoon Washi. However, the main goal of the study was to determine whether traumatic memories predicted negative mental disorder outcomes, and the authors did not directly report mental disorder prevalence estimates in this case. We chose to include these types of studies because they met our eligibility criteria. In addition, with the overall lack of literature on this topic and our goal to create an extensive map of the research, we felt that they were important to include. We noted this in the discussion section (lines 326-330). |
Round 2
Reviewer 2 Report
The paper is much inproved.
In the second draft the focus is on the relationship between natural disaster and mental disorder, but not the factors affecting this relationship. I perceived that as a missed opportunity (as I indicated in my first review). Especially since the research question focusses of how exposure to climate-related disasters influences the presence of mental disorder. How implies a process, not only a relationship. The process could be captured by looking at other variables that affect the relationship, that can be summarized in a conceptual model. As mentioned in the discussion, some papers use a conceptual framework, so why not use that information to make a theoretical step forward.
Reviewer 3 Report
No extra comments or suggestions.